# Control of Plant Viral Diseases by CRISPR/Cas9: Resistance Mechanisms, Strategies and Challenges in Food Crops

**DOI:** 10.3390/plants10071264

**Published:** 2021-06-22

**Authors:** Saleh Ahmed Shahriar, M. Nazrul Islam, Charles Ng Wai Chun, Md. Abdur Rahim, Narayan Chandra Paul, Jasim Uddain, Shafiquzzaman Siddiquee

**Affiliations:** 1School of Biological Sciences, Universiti Sains Malaysia, Penang 11800, Malaysia; shahriar@student.usm.my; 2Laboratory of Plant Pathology and Microbiology, Morden Research and Development Centre, Agriculture and Agri-Food Canada, Morden, MB R6M 1Y5, Canada; naz.islam@canada.ca; 3Bioprocess Technology Division, School of Industrial Technology, Universiti Sains Malaysia, Penang 11800, Malaysia; charlesngwaichun@student.usm.my; 4Department of Genetics and Plant Breeding, Sher-e-Bangla Agricultural University, Dhaka 1207, Bangladesh; rahimgepb@sau.edu.bd; 5Department of Integrative Food, Bioscience and Biotechnology, Chonnam National University, Gwangju 61186, Korea; ncpaulbd@jnu.ac.kr; 6Department of Horticulture, Sher-e-Bangla Agricultural University, Dhaka 1207, Bangladesh; jasimhort@sau.edu.bd; 7Biotechnology Research Institute, Universiti Malaysia Sabah, Jln UMS, Kota Kinabalu 88400, Malaysia

**Keywords:** CRISPR/Cas9, food crops, plant viral diseases, resistance mechanisms and strategies, challenges

## Abstract

Protecting food crops from viral pathogens is a significant challenge for agriculture. An integral approach to genome-editing, known as CRISPR/Cas9 (clustered regularly interspaced short palindromic repeats and CRISPR associated protein 9), is used to produce virus-resistant cultivars. The CRISPR/Cas9 tool is an essential part of modern plant breeding due to its attractive features. Advances in plant breeding programs due to the incorporation of Cas9 have enabled the development of cultivars with heritable resistance to plant viruses. The resistance to viral DNA and RNA is generally provided using the Cas9 endonuclease and sgRNAs (single-guide RNAs) complex, targeting particular virus and host plant genomes by interrupting the viral cleavage or altering the plant host genome, thus reducing the replication ability of the virus. In this review, the CRISPR/Cas9 system and its application to staple food crops resistance against several destructive plant viruses are briefly described. We outline the key findings of recent Cas9 applications, including enhanced virus resistance, genetic mechanisms, research strategies, and challenges in economically important and globally cultivated food crop species. The research outcome of this emerging molecular technology can extend the development of agriculture and food security. We also describe the information gaps and address the unanswered concerns relating to plant viral resistance mediated by CRISPR/Cas9.

## 1. Introduction

Food crops are vulnerable to various diseases including bacteria, fungi and viruses, resulting in major economic losses. Thus, crop resistance can be improved against various pathogens [1,2]. Disease control approaches rely on resistant cultivars and agrochemicals that are typically highly effective once deployed. Protecting crops from emerging pests and diseases and developing resistant cultivars from the perspective of higher production are significant challenges [3]. Because pesticides used in the field are usually not highly selective, they can also affect other beneficial organisms when killing pathogens, and thus disrupt the ecological balance. The development of disease-resistant crop varieties is an efficient and environmentally responsible integrated agriculture policy [4]. Various genome editing and sophisticated molecular technologies for different transgenic crop plants are combined in modern plant breeding. Thus, improved crop varieties can be obtained with enhanced disease resistance, which is known as resistance plant breeding. This transgenic technology enables plant breeders to crossbreed crop species and insert desired genes from non-related species and/or organisms into crop plants [5].

Genetic diversity with enhanced virus protection is an indispensable part of virus resistance [6]. New breeding techniques (NBTs) comprise the latest and most effective biological methods for the accurate genetic manipulation of single or several target genes. They use a site-driven nuclease to add double-stranded breaks in DNA at specified regions. These breaks are usually repaired through various host-genome cell repair mechanisms, thus resulting in either small insertions or deletions via non-homologous end-joining (NHEJ) or a modified gene carrying the predetermined nucleotide changes copied from a repair matrix via homologous recombination (HR) [7]. CRISPR/Cas9 system highlight the reality that this technique requires fewer skills and financial resources and has a higher accuracy rate for gene editing performance, as in the Cas9 system, crops require off-target cleavage and different goal choices relative to the other nucleases available [8].

CRISPR/Cas9 is considered a highly promising genome-editing method in crops due to its unique features, such as reliable precision, multiple-gene editing, limited off-target impact, greater output, and simplicity [9]. This particular mechanism invades foreign DNA fragments of virus particles and enables them to detect and degrade the DNA or RNA sequences for further invasion [10]. CRISPR/Cas9 technology manipulates the defense mechanism against plant viruses by identifying and destroying pathogenic genes that invade them. It can also be deployed to develop crop cultivars with enhanced resistance against various plant viruses. This approach has revolutionized research in virus resistance due to its sequence-specific nuclease capability [11]. The use of association genetics focused on single nucleotide polymorphisms (SNPs) and other common molecular markers has also increased in plant breeding, generating important high throughput data for quantitative trait loci (QTL) recognition. The major QTL has been used in crop varieties to provide the quantitative resistance to plant viruses, combined with the usage of primary resistance (*R*) genes incorporated into cultivars with superior agronomic characteristics [12]. In this review, we address the key features of the CRISPR/Cas9 genome-editing technique and its implications for new food crop cultivar evolutions with improved plant virus resistance.

## 2. Virus-Resistance Mechanism of CRISPR/Cas9 in Crops

The CRISPR/Cas9 method identifies and targets a pathogen’s genetic material via a three-step procedure: (i) acquisition, (ii) expression and (iii) interference [13]. The first step, acquisition, involves the invasive foreign DNA as a spacer from viruses or plasmids, which are divided into short fragments, and recognized and incorporated into the CRISPR locus. CRISPR loci have been copied and used to produce short RNA (crRNA), directing the effector endonuclease genes through simple complementarity to target the virus components. In general, the protospacer adjacent motif (PAM) includes a short stretch (2–5 bp) of retained nucleotides that obtain the DNA fragment (spacer) for identification. Mutations in viral genomes and PAM overcome the CRISPR-mediated immunity from pathogen attacks (Figure 1) [14].

CRISPR/Cas9 expression includes effectively transcribing the large pre-CRISPR RNA (pre-crRNA) attained from the CRISPR locus, thus converting it into several crRNAs with the aid of the Cas9 protein and the tracrRNA molecule [15]. The tracrRNA combines with the crRNA repeat area through base complementarity and facilitates pre-crRNA processing in crRNA (Figure 2) [16]. The activated crRNAs join the CRISPR-associated antiviral protection complex (CASCADE) and help to identify and base-pair a particular target area of foreign DNA [17].

In the CRISPR/Cas9 method, DNA interference requires a single Cas9 protein [18]. The crRNA guides the Cas9 protein into the target site of the foreign DNA to break down during the interference step and provides immunity against pathogen attacks [13]. Cas9 is an immense protein with several domains (the amino terminus RuvC domain and the centrally-located HNH nuclease domain) and two short RNA segments, namely, crRNA and trans-activating crRNA (tracrRNA). The Cas9 protein promotes adaptation, engages in pre-crRNA processing toward crRNA, and implements specified DNA double-strand breaks (DSBs) led by the tracrRNA and RNAse III-specific double-stranded RNA [19]. The creation and development of CRISPR/Cas9 structures are relatively simple, affordable, and without intellectual property obstacles. The CRISPR/Cas9 tool, crRNA, and tracrRNA components can be fused together into the sgRNA that guides Cas9 to implement the targeting of specific DSBs (Figure 2) [16]. The design of sgRNAs is also straightforward, thus favouring genome-editing. The CRISPR/Cas9 method was initially designed to cause cleavage in DNA in vitro at different sites [19,20]. This method has recently been implemented to edit genomes in bacteria, fungi, viruses, yeast and many other organisms for successful selective mutagenesis [20,21,22,23].

## 3. Plant Virus-Resistance Strategies Using Cas9 Endonuclease

The CRISPR/Cas9 technology allows for the development of a broader range of CRISPR variants useful for different applications and has been successfully demonstrated to engineer virus-resistant crop cultivars. However, gene disruption is one of the most common applications of the CRISPR/Cas9 tool (Figure 3) [24] and helps to overcome the error-prone behaviour of cellular NHEJ (DNA-repair machinery). The insertion/deletion (InDel) of nucleotides at sgRNA-targeted sites introduces a frameshift mutation and disrupts gene function [25]. In the context of virus resistance, this strategy has been employed to engineer resistance by disrupting the susceptible (*S*) gene(s) function, which alters the plant–virus interaction, resulting in reduced viral fitness in the host plant. Another approach can also be utilized to introduce InDels in the promoter region rather than the coding region of a gene [24]. CRISPR-mediated promoter disruption blocks the entire gene expression and the effector-binding site of susceptible plants by preventing a virus effector from binding to the promoter [26]. Moreover, due to the gene clusters, it is feasible to use this particular approach to engineer virus resistance in plants. Deleting the chromosomal fragments adjacent to *S* gene clusters may produce durable virus resistance in different hosts. In addition to virus resistance, CRISPR-mediated gene insertion opens an avenue to study important *S* genes [27]. The functional analysis of susceptibility genes enables us to understand the regulation of gene expression. The host proteins play key roles in the pathogenicity of the genome, resulting in their expression and localization during viral infection. Recent studies have recognized several resistance (*R*) gene(s) in wild species and proved the successful transfer of resistance in cultivated crop species [25]. This tool can replace the improper and poorly performing *R* genes from cultivated crop species with a functional *R* gene from a virus-resistant cultivar via multiplexed homology directed repair (HDR) methodology. The biomimicry mechanism introduces the CRISPR-mediated mutations, which convert the target gene as the sequence of a virus-resistant cultivar [26]. This approach is beneficial to introduce only specific mutations associated with virus resistance traits instead of replacing the whole gene (Figure 3) [24]. In this manner, it can be assumed that the differences in the nucleotide sequences of the target genes among the cultivated species and wild varieties are not sufficiently significant to confer durable resistance to various plant viruses. In this case, the Cas9 tool can be utilized to modify a particular gene for stable plant resistance. The durability of resistance primarily relies on the appropriate molecular strategies within Cas9 mechanisms that are applied. 

## 4. Deletion and Insertion of the Target Gene

Multiple sgRNA sequences can be utilized to drive the numerous double-strand breaks (DSBs) through CRISPR/Cas9 at the appropriate site of the target gene. Two sgRNA bindings will generate two DSBs at the respective sites before the start codon and after the stop codon of the gene of interest. These DSBs then remove the DNA fragments carrying the gene of interest before repairing the cellular non-homologous end-joining (NHEJ) machinery (Figure 4) [28]. The sgRNA bindings can be designed for any gene region carrying a precise trinucleotide protospacer adjacent motif (PAM). This approach deletes the expanded chromosomal fragments, including the individual genes [29,30].

The CRISPR/Cas9 tool can be employed to enhance disease resistance via alteration of the susceptible (*S*) genes. These *S* genes disrupt the plant proteins, which are essential and primarily multifunctional. Thus, plant health and productivity are also retarded. An alternate approach, such as cis-regulatory component and promoter editing, can alter gene expression instead of gene disruption. It is sometimes crucial to utilize resistance (*R*) genes against virulent pathogens. In contrast, the host–pathogen interaction has not been fitted well, and the *S* genes have not been explored extensively [31]. In such cases, the Cas9 technique can be utilized for the insertion of the *R* genes. Insertion of the CRISPR-mediated gene operates through an alternative pathway that works when Cas9 generates the sgRNA-directed DSBs; this pathway uses cellular homology-directed repair (HDR) instead of non-homologous end-joining (NHEJ) machinery (Figure 4) [28]. The delivery fragment carrying an *R* gene, which is surrounded by the homologous sequence of DSB ends, is adjunct with the Cas9-gRNA complex. This cassette directs the insertion of the HDR-mediated R gene between the two DSB ends. This approach was practiced to introduce many genes at the distinct genomic regions [32].

## 5. Off-Target Mutations in CRISPR/Cas9

The CRISPR/Cas9 technology evaluates the high specificity in targeting the genomic regions, such as irradiation-induced mutagenesis. This tool raises questions about how a sgRNA can target the entirely complementary genomic DNA sequences and other genomic sites (off-target regions) and the extent of targeting and potentially provoke unexpected damage. There are two types of expected off-target effects in genomic sites: higher sequence similarities to the target and the unexpected off-target in irrelevant genomic sites. Gene sequence information is essential to predict the expected off-target outcomes [25]. Random off-target mutations have become more frequent when mismatches occurred from distant genomic regions [33]. Recent studies have screened polyploidy progenies via CRISPR/Cas9 knockout to clarify the off-target issue in crops. This may bind the lower effective sequences with the mismatches (usually 1 to 3). The appropriate design of the CRISPR/Cas9 tool can avoid the expected off-target mutations, but unexpected off-target mutations cannot be avoided because of their spontaneous lower frequency mutations of plants [27].

## 6. Overcoming Off-Target Effects

To enhance the specificity and effectiveness of the CRISPR/Cas9 system, it is vital to analyze and address its possible off-target mutations. Off-target effects generated by the CRISPR/Cas9 system can be identified in plants via qRT-PCR resequencing [34]. The off-target effects have become a potential concern for the CRISPR/Cas9 system. However, few studies have addressed this issue in various organisms. Recently, the high-resolution structure of the SpCas9–sgRNA complex has removed the mismatch tolerance mechanism between the sgRNAs and the targeted DNA sequences [35]. This finding can be considered a significant advance towards understanding the specificity and molecular mechanism of the CRISPR/Cas9 system. To overcome the off-target effects, a mutant Cas9 endonuclease was used to induce the distinct cleavage only at one DNA strand of the targeted site and activate the homologous recombination to enhance the specificity of the target region [27]. The Cas9-sgRNA complex reduced off-target mutations by 50- to 1500-fold by introducing DSBs. The significant numbers of Cas9/sgRNA delivered, and the ratio of Cas9:sgRNA, have also been found to minimize off-target effects. A higher concentration of the Cas9:sgRNA ratio results in higher off-target rates than a lower Cas9/sgRNA concentration [25].

## 7. CRISPR/Cas9 Toolkit in Crop Improvement

Due to its potentiality, low cost, and simplicity, Cas9 has become a widely adopted genome-editing tool in numerous living organisms, including plant species. Editing genes allow the contemporaneous alteration of multiple genetic loci, thus accelerating various commercial crop improvements and enhancing food safety globally [36]. A number of recent studies using this technique for genome-editing have focused on different economically important food crops necessary for modern agriculture. It has been found that this technique could be utilized for improvement of specific traits, such as crop yield, grain quality and disease resistance [37]. CRISPR/Cas9 has been practiced in the genome-editing of rice (*Oryzae sativa*). Several Cas9-sgRNAs were designed to effectively delete the small fragments of the *dense and erect panicle1* (*DEP1*) gene in the Indica rice line. These improvements in traits that contribute to yield, such as reduced plant height and dense and upright panicles, have been executed in the production of mutant plants [38]. This method was utilized to introduce mutations into the *GmFT2a* gene, an essential integrator in photoperiodism of the flowering pathway of soybean plants. The newly developed soybean cultivars exhibited late flowering, which results in the increased size of the vegetative growth. This mutation was also firmly inherited among the following generations [39]. CRISPR/Cas9 was used to generate mutations in the flowering suppressor gene *SELF-PRUNING5G* (*SP5G*) in tomato plants to manipulate the photoperiod response. These mutations were achieved using Cas9 and caused rapid flowering and augmented the dense growth habit of tomato plants, resulting in an immediate increase in early yield [40]. The development of resistance against *Xanthomonas citri* subsp. *citri*, which causes bacterial citrus canker, was undertaken using Cas9 technology. A promoter gene *CsLOB1* was targeted, which promotes the development of canker in citrus plants. The developed transgenic lines exhibited improved resistance against canker in citrus compared to the wild types [41]. Bread wheat cultivars mutated using the CRISPR/Cas9 toolkit exhibited improved resistance to infection with *Blumeria graminis* f. sp. *tritici*. This finding also showed the development of powdery mildew resistance in wheat [42].

## 8. Genetics of Plant Virus Resistance

Although viruses are relatively simple genetic entities, many of their mechanisms by which diseases are generated, and by which plants can resist these effects by natural resistance, remain unknown [43]. The most effective and sustainable approach to protect plants from virulent viruses is the development of genetic resistance against these viruses. The rigorous study of plant resistance (*R*) genes, in which genetic variability occurs that alters the plant’s suitability, raises numerous fundamental questions regarding the molecular, biochemical, cellular and physiological mechanisms involved in the plant–virus interaction [5]. Recently, significant advances have been made in molecular mechanisms associated with natural virus resistance genes. Both the dominant and recessive resistance genes have been characterized at the molecular level to understand new principles of innate immunity to viruses associated with gene silencing. Thus, it has been possible to undertake genome sequencing of many crop species, and this technology is now being utilized on a larger scale. These advances also provide new opportunities to tackle the barriers to virus resistance [3]. Resistance breeding should not depend on extreme molecular characterization of resistance gene alleles and the target a virulence (avr) determinant of a virus. In a practical sense, the successful deployment of a potential resistance gene into a crop depends more upon the identification of a positive phenotype, on the dissection of the phenotype, leading to the identification of genetic markers for marker-assisted selective breeding (MAS) and on an understanding of how the novel resistance will behave in different genetic backgrounds and under pathogen pressure in the field [44]. However, the focus is mainly on monogenic dominant resistance to bacterial and fungal pathogens, and the common mechanisms can also be employed in virus resistance. Low resistance durability is a barrier to the appearance and increased frequency of resistance-breaking (RB) variants among virus populations. The genetic changes required for a virus to overcome plant-resistance mechanisms and the effects of these changes on the fitness of the virus are key determinants of resistance durability.

## 9. Economic Importance of Plant Viral Diseases in Food Crops

Plant viruses have immense importance in agriculture because many infect different food crops such as vegetables, fruits and staple grains, causing a deterioration in productivity and quality. One virus may infect several crop species, and different kinds of viruses usually attack each species of crop. Viruses cause disease by consuming cells or killing them with toxins and utilizing cellular substances during multiplication, taking up space in cells and disrupting cellular processes. Almost all viral diseases appear to cause dwarfing or stunting of the entire crop plant and a reduction in the total yield. These effects may be severe and cause striking symptoms on the stem, fruit and roots; alternatively, they may or may not cause any symptom development. Plants may show acute severe symptoms soon after inoculation that may lead to the death of young shoots or the entire host plant. If the host survives the initial shock phase, the symptoms tend to become milder (chronic symptoms) in the subsequently developing parts of the plant, leading to partial or even total recovery.

However, in some diseases, symptoms may increase progressively in severity and may result in a gradual or quick decline in the health of the plant. The most common types of plant symptoms produced by systemic virus infections are mosaics and ring spots. Mosaics are characterized by light-green, yellow or white areas with the typical green color of the leaves, flowers and fruits. Depending on the intensity or pattern of discolorations, mosaic symptoms may be described as mottling, streaks, a ring pattern, a line pattern, vein clearing, vein banding or chlorotic spotting. Ring spots are characterized by chlorotic or necrotic rings on the leaves and sometimes also on the fruits and stems. In many ring spot diseases, the symptoms tend to disappear later, but the virus remains. The extent of economic yield loss due to viral infection in many crop varieties is summarized in Table 1 and Table 2 present specific examples of key resistance techniques by CRISPR/Cas9 used in several food crops from 2015 to 2021. 

## 10. Challenges

The CRISPR/Cas9 platform is a groundbreaking advancement that allows the engineering of interference against plant pathogens. In-depth research is required to completely develop the effectiveness of this method to globally combat hunger due to crop losses caused by viral diseases. One of the concerns of the CRISPR/Cas9 tool is the risk of newly emerging recombinant viruses that are resistant to the Cas9 endonuclease [59]. These recombinant viruses are capable of escaping CRISPR/Cas9 targeting due to few InDels target of the sgRNA sites. These mutants are not lethal to viral replication, but they impede the sgRNA sites from identifying the targeted genomic sequence encompassing the Cas9 endonuclease [25]. The mutation and recombination of endogenous genes may lead to their functional inactivation, whereas the variations of virus sequences produced by CRISPR/Cas9 can facilitate viral evolution. It is well known that mutation and recombination are the major driving forces of plant viruses [60].

CRISPR/Cas9 targeting viral open reading frames (ORFs) can generate more viral variants, resulting from the viral escape events at different levels. Cas9 may disseminate these mutant viruses from plants and with multiple sgRNAs [26]. In addition to targeting or interfering with the viral genome to inhibit its infection, CRISPR/Cas9 can generate new viral variants as genome-editing by-products that speed up the evolution of the virus, causing the produced transgenic crops to lose their resistance capacity against these viruses [44]. These limitations are shortcomings of the CRISPR/Cas9 tool against some plant viruses. Several molecular approaches can be used to combat these recombinant viral escapes; for instance, utilizing multiplex genome engineering to target the different sites of the virus genome can significantly inhibit the virus replication [58].

A number of reports were found regarding resistance genes in many economically important food crops, including the 20 viruses listed in Table 3. Cas9 technology could be a potential genomic tool to determine the host–virus interaction mechanisms with the listed genes (Table 3). Currently, these viruses are controlled using chemical management, which has little or no effectiveness. In addition, synthetic inorganic chemicals are toxic to soil and water and are harmful to health. Alarmingly, hazardous chemical molecules continue to recycle within the agroecosystems.

## 11. Research Opportunities

This review highlighted that CRISPR/Cas9 technology has been used to develop crop varieties resistant to numerous significant viral pathogens. Furthermore, this technique may also provide an important opportunity to research these viruses, resulting in benefits to humans. Viruses cause numerous diseases in the most economically significant crop plants, and CRISPR-mediated virus resistance has been used to target the virus DNA and RNA. This approach has some limitations; notably, viruses have the capacity to escape and generate resistance-blocking strains. Specific utilization of the host susceptibility factors involved in plant–virus interactions is a possible solution for this shortcoming. Moreover, the well-distinguished susceptible (*S*) gene(s) available for other pathogens are usually absent for viruses. Considerable research has been conducted to identify and understand the mechanisms of virus infection, and fewer potentially relevant plant genes have also been recognized. These comprehensive reviews show that the host susceptibility factors may potentially target the S genes to develop resistance against the crop virus, which is a promising future approach for agriculture. The mutation among the plants generated using the CRISPR/Cas9 method is usually stable and heritable in nature. This mutation can be easily separated from the Cas9/sgRNA complex to avoid further alterations by Cas9/sgRNA, thus helping to develop transgene-free progeny in only one generation. The integration of Cas9/sgRNA in the CRISPR/Cas9 toolkit for virus resistance assists in the development of virus-resistant transgenic plants. The plasmid-free delivery of pre-assembled complexes is another essential technique for generating virus-resistant plants that contain no foreign DNA in the respective virus genome. Identifying potential genes or DNA sequences that can be utilized as useful targets for gene editing is also essential. For example, the fission of preserved non-coding regions with the virus genome contributes to the same taxonomic community’s resistance to specific viruses. Using the prokaryotic CRISPR/Cas9 framework means that the resistance produced would be more effective because, unlike the RNA interference method, viruses lack the machinery to prevent their inhibitory effects. The Cas9 protein and short-range in vitro transcribed RNA guide can be incorporated directly into the cells as a ribonucleo protein complex (RNP), which does not require the integration of transgenes into the genome. These approaches apply to RNA or DNA genome viruses. In addition, the rate of recombination and mutation of viral DNA also remains to be explored immediately after the cleavage virus genome enters the host cell. Among other questions, it is also unknown whether the multiplexing of sgRNAs in transgenic plants may promote resistance to diverse strains of plant viruses, because it is naturally covered with the crRNA gene within the prokaryotic CRISPR arrays.

The presence of off-target results continues to be investigated using CRISPR/Cas9 in vivo conditions among the individual hosts. In the CRISPR/Cas9 method, the target RNA binds the DNA, and pre-designed sequences remaining inside the RNA lead the Cas9 protein to break DNA strands at the correct positions. DNA cutting is achieved by extracting and adding the appropriate sequences of the target DNA. Homologous recombination of DNA restores the DNA damage and causes mutations. However, it also often contributes to unintended mutations because it is not appropriate to complement a total of 20 base pairs in the short-range RNA guide sequence to bind the target DNA. This may also result in the attachment of lead RNA in an inappropriate location and cutting of the non-targeted DNA, resulting in improper mutations and perhaps causing the loss of the essential segment of DNA sequences with necessary details. However, research is being undertaken to allow more practical application of CRISPR/Cas9 because the Cas9 protein can be managed to target specific DNA strands by modifying the sequences of the RNA guide.

## 12. Conclusions

The CRISPR/Cas9 method shows promise as a useful genome-editing technology for producing different virus-resistant crop cultivars with improved yield, quality, and abiotic/biotic stress tolerance. The off-target effects are among the limitations for large-scale field application of the CRISPR/Cas9 tool in virus-resistant crop development. This challenge may be overcome by applying more effort to the machine learning (ML) process. At present, the impact of the CRISPR/Cas9 methodology on plant physiological traits and potential virus mutants is inadequately documented. Global collaboration using open field trial data (from multiple climates and locations) on Cas9-treated crop plant–virus–environment interactions, and the subsequent growth and yield performance of staple food crops, warrants further attention.

## Figures and Tables

**Figure 1 plants-10-01264-f001:**
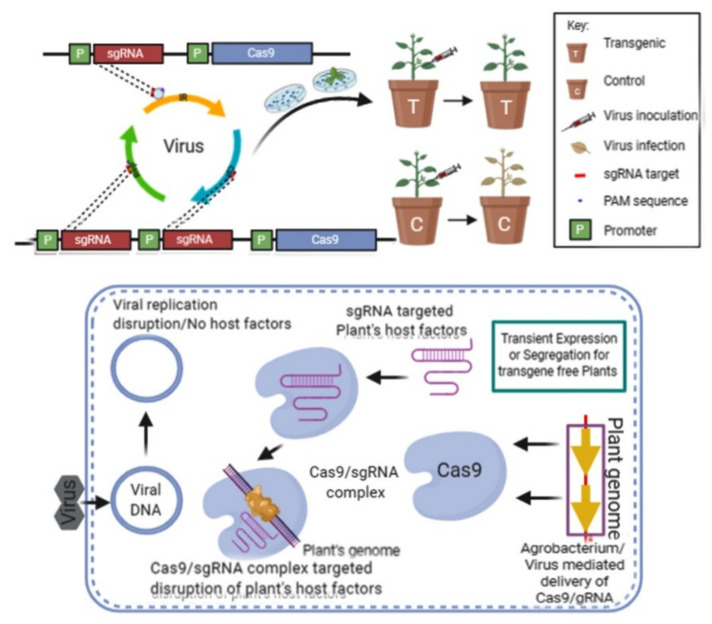
Virus resistance via CRISPR/Cas9 (illustration is adapted and modified from Zaidi et al. [14]).

**Figure 2 plants-10-01264-f002:**
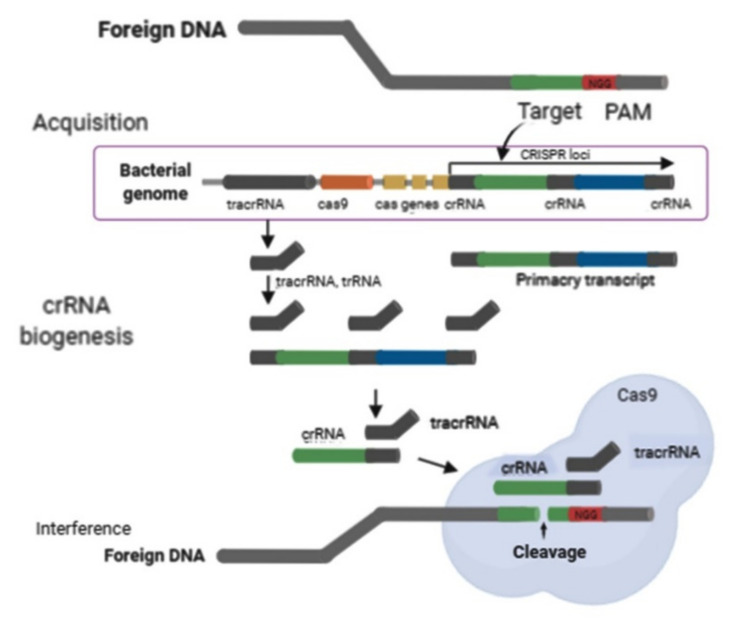
Mechanism of CRISPR/Cas9 system (illustration is adapted and modified from Arora et al. [16]).

**Figure 3 plants-10-01264-f003:**
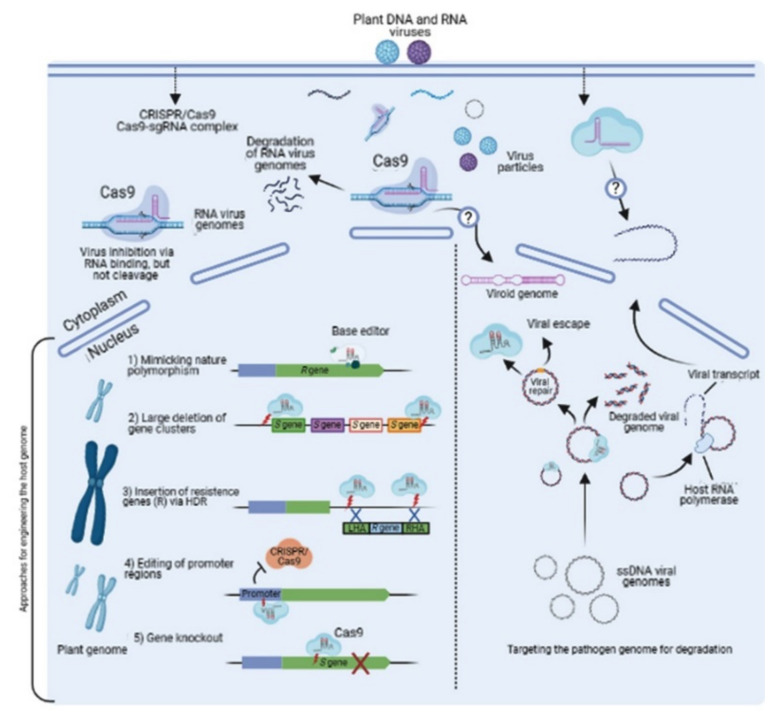
Plant virus-resistance strategies using Cas9 (illustration is adapted and modified from Zaidi et al. [24]).

**Figure 4 plants-10-01264-f004:**
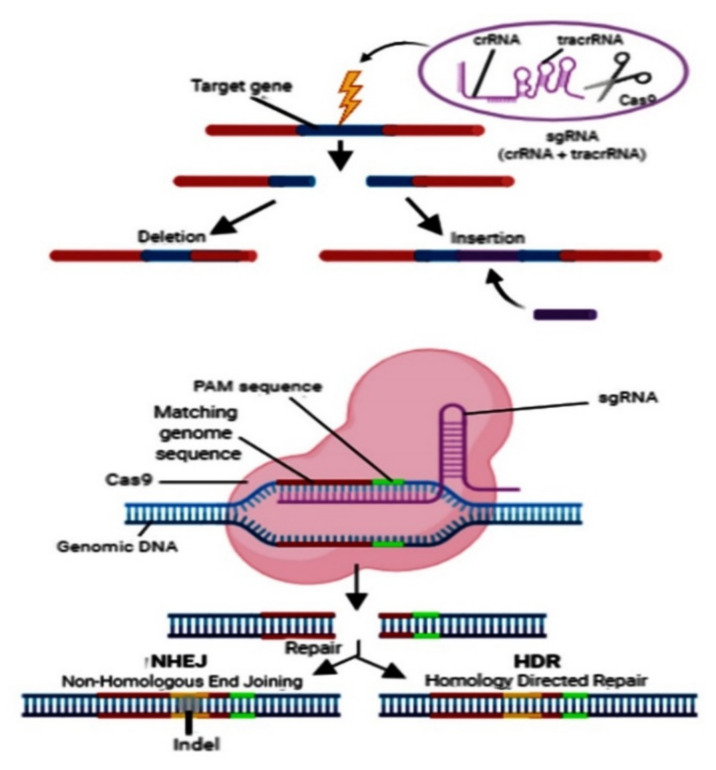
The deletion and insertion of the target gene (illustration is adapted and modified from Ghosh et al. [28].

**Table 1 plants-10-01264-t001:** A list of viruses causing diseases in food crops via insect vectors and estimated yield loss in different regions of the world; results summarized from G. N. Agrios (2005) [45].

Virus	Virus Family	Damaging Food Crop	Distribution	Vector	Crop Yield Loss
African cassava mosaic virus (ACMV)	Geminiviridae	Cassava	Africa and all cassava growing countries	Whitefly	20–90%
Banana bunchy top virus (BBTV)	Nanoviridae	Banana	All banana growing countries	Aphid	Up to 100%
Banana streak virus (BSV)	Caulimoviridae	Banana	All banana growing countries	Mealybug	6–15%
Barley yellow dwarf virus (BYDV)	Luteoviridae	Barley, oats, rye and wheat	Global	Aphid	30–50%,
Bean yellow dwarf virus (BeYDV)	Potyviridae	Bean, peas and other legumes and non-legumes	Global	Aphid and mechanical infection	30–70%
Bean common mosaic virus (BCMV) and bean yellow mosaic virus (BYMV)	Potyviridae	BCMV infects the only bean, BYMV also infects peas and yellow summer squash	Global	Aphid and seeds from infected plants	Up to 100%
Bean golden mosaic virus (BGMV)	Geminviridae	Bean, other legume crops and Malvaceous weeds	Global	Whitefly	Up to 100%
Beet severe curly top virus (BSCTV)	Geminiviridae	Sugar beet, bean, melons, spinach, sugar beet and tomato	Global	Leafhopper	Up to 100%
Beet yellows virus (BYV)	Closteroviridae	Sugar beets, spinach and table beets	All sugar beet growing countries	Aphid	Up to 50%
Cassava brown streak virus (CBSV)	Potyviridae	Cassava	East Africa including Kenya, Mozambique and Tanzania	Whitefly	Up to 30%, reduced market price up to 90%
Cauliflower mosaic virus (CaMV)	Caulimoviridae	Vegetables	Many parts of the world	Aphid	20–50%
Cotton leaf curl kokhran virus (CLCuKoV)	Geminiviridae	Okra	China, India, Pakistan and Philippines	Whitefly	Up to 55%
Cucumber mosaic virus (CMV)	Bromoviridae	Bananas, beans, beets, celery, crucifers, cucumbers, melons, peppers, spinach, squash, tomatoes and vegetables	Global	Aphid	80% and above
Cucumber vein yellowing virus (CVYV)	Potyviridae	Cucumber, melon or watermelon and other crops of Cucurbitaceae family	Global	Whitefly	Up to 70%
Grapevine fan leaf virus (GFLV)	Secoviridae	Grape	Global	Nematodes	Up to 80%
Lettuce infectious yellows virus (LIYV)	Closteroviridae	Lettuce, cantaloupe, carrot, melon, squash, sugar beet, squash	Global	Whitefly	30–100%
Lettuce mosaic virus (LMV)	Potyviridae	Lettuce, marigold, pea and sweet pea	Global	Aphid and infected seeds	55–85%
Maize dwarf mosaic virus (MDMV)	Potyviridae	Maize	Global	Aphid	Up to 70%
Maize streak virus (MSV)	Geminiviridae	Maize, rice, wheat, millet and sugarcane	India and southern part of Africa	Leafhopper	Up to 100%
Merremia mosaic virus (MeMV)	Geminiviridae	Hot pepper, sweet pepper and so on.	Where irrigated crop production practiced worldwide	Whitefly	70–80%
Okra yellow vein mosaic virus (OYVMV)	Geminiviridae	Okra	Global	Whitefly	More than 90%
Papaya ring spot virus-W (PRSV-W)	Potyviridae	Papaya and cucurbits	Global	Aphid	Up to 100%
Plum pox virus (PPV)	Potyviridae	Apricot, nectarine, peach and plum	Global including Asia, Europe and North America	Aphid, budding and grafting	80–100%
Potato leafroll virus (PLRV)	Luteoviridae	Potato	Global	Aphid	Up to 90%
Potato virus X (PVX)	Alphaflexiviridae	Potato, pepper, tobacco and tomato	Global	Handling of the plant materials	30–40%
Potato virus Y (PVY)	Potyviridae	Potato, pepper, tobacco and tomato	Global	Aphid	Up to 70%
Rice tungro bacilliform virus (RTBV) and Rice tungro spherical virus (RTSV)	Caulimoviridae (RTBV), Secoviridae (RTSV)	Rice	South and Southeast Asia	Leafhopper	Up to 100%
Soybean mosaic virus (SMV)	Potyviridae	Soybean and various crops of Fabaceae and Leguminosae family	Global	Aphid	35–90%
Squash leaf curl virus (SLCV)	Geminiviridae	All cucurbits such as squash, cucumber, watermelon and cantaloupe	Southern region of California	Whitefly	70–80%
Sugarcane mosaic virus (SCMV)	Potyviridae	Corn, sorghum, sugarcane and other crops of Gramineae family	Global	Aphid	Up to 40%
Tomato mosaic virus (ToMV)	Virgaviridae	Tomato and other Solanaceous crops	Global	Infected seeds	20–90%
Tomato mottle virus (TMoV)	Geminiviridae	Common bean, tobacco and tomato	Global	Whitefly	Up to 95%
Tomato ring spot virus (TomRSV)	Secoviridae	Tomato, strawberries, raspberries, grapes and apple	Global	Nematodes	50–80%
Tomato spotted wilt virus (TSWV)	Bunyaviridae	Lettuce, papaya, peanut, pineapple, tomato and various fruits and vegetables	Tropics and subtropics and a few temperate regions of the world	Thrip	50–90%
Tomato yellow leaf curl Sardinia virus (TYLCSV)	Geminiviridae	Watermelon, tomato, squash, potato, pepper, melon, cotton, cassava and bean	Global particularly prevalent in tropics and subtropics regions	Whitefly	Up to 100%
Tomato yellow leaf curl virus (TYLCV)	Geminiviridae	Tomato and many food crops	Global	Whitefly	Up to 100%
Turnip mosaic virus (TuMV)	Potyviridae	Brussels, sprouts, cabbage and cauliflower	Global	Aphid	Up to 70%
Watermelon mosaic virus (WMV)	Potyviridae	Peas and Leguminous crops	Global	Aphid	Up to 100%
Wheat dwarf virus (WDV)	Geminiviridae	Wheat and barley	Throughout Europe	Leafhopper	Up to 75%
Zucchini yellow mosaic virus (ZYMV)	Potyviridae	Cucumber, muskmelon, watermelon, zucchini squash.	Global	Aphid	Up to 70%

**Table 2 plants-10-01264-t002:** A meta-analysis of CRISPR/Cas9 strategies used for virus resistance in selected food crops; data summarized from 2015–2021 published studies.

Virus in Food Crops	Experimental/Model Host	Targeted Gene Region(s)	Key Strategies of CRISPR/Cas9	Reference
Banana streak virus (BSV)	*Arabidopsis thaliana*	BSOLV and eBSOLV	The gRNAs were designed to target the sequences of BSOLV and eBSOLV via the CRISPR/Cas9 technology. Three gRNAs based on their specificity to their target site (targeting sequence S1, S2 and S3 from ORF1, ORF2 and ORF3, respectively) and minimal potential off-targets were introduced into the triploid Musa genome (Gonja Manjaya, AAB). The gRNA cassette, containing OsU6 promoter followed by two BbsI restriction sites and tracer RNA scaffold, was amplified from pZKOsU6-gRNA plasmid and cloned into pENTR-D/Topo. One gRNA from each ORF of the BSV genomic sequence was synthesized and cloned into pMR185 to generate the gRNA modules. The Cas9 endonuclease was employed in the plasmid of *Arabidopsis* codon-optimized and regulated by parsley ubiquitin promoter (PcUbi).	[46]
Bean yellow dwarf virus (BeYDV)	*A. thaliana*; *Nicotiana benthamiana*	IR, CP, and Rep protein	CRISPR/Cas9 technology was successfully utilized in engineering resistance to the bean yellow dwarf virus (BeYDF). Baltes et al. [46] established a transient assay to detect the activity of Cas9 and sgRNA in *N. benthamiana* using BeYDV. They used double 35S promoter and AtU6/At7SL RNA polymerase III promoter to express Cas9 and sgRNAs, respectively. The enhanced green fluorescent protein (eGFP) gene replaced the coat protein genes and movement protein for Cas9 and sgRNAs activity assessment against BeYDV in *N. benthamiana*.	[47]
Beet severe curly top virus (BSCTV)	*N. bethamiana*	CP and Rep protein	The CRISPR/Cas9 tool constructed two vectors (pV86-401 and pC86-401) in which the Cas9 protein is driven by one or two BSCTV promoters and sgRNA complex is driven by AtU6 promoter. The expression of Cas9 under both pV86 and pC86 promoters was significantly induced after BSCTV accumulation in *N. benthamiana* and pC86 promoter appeared to enable higher-level induction than the pV86 promoter. Previously employed pV86-sgRNA and pC86-sgRNA vectors using four highly active sgRNAs were then constructed to determine the system efficiency against BSCTV.	[48]
Cassava brown streak virus (CBSV)	*Morchella esculenta*	eIF4E, *nCBP-1* and *nCBP-2*	Cassava brown streak virus (CBSV) is a significant threat to cassava production in Africa. For the disease development in the host, CBSV requires the interaction between “viral genome-linked protein (VPg)” and “eukaryotic translation initiation factor 4E (eIF4E) isoforms” of the host. The nCBP clade was consistently associated with VPg protein and given priority due to its functional characterization. The CRISPR/Cas9 system was employed to produce mutant alleles of nCBP isoforms in cassava. Five constructs were combined, simultaneously targeting the different locations within nCBP-1 and nCBP-2 clades.	[49]
Cauliflower mosaic virus (CaMV)	*A. thaliana*	CP	The target sites in the CaMV CP gene were selected using standard Cas9 protein. Linear arrays of Arabidopsis U6 promoter: sgRNA units were designed and subsequently synthesized. When controlling viruses using the CRISPR-Cas9 system, both Cas9 and sgRNAs are consistently expressed in the cells. Recruiting Cas9 to viral DNA depends on the presence and abundance of sgRNAs. However, due to the existence of folded dsRNA domains in sgRNAs, siRNAs can be formed to contain the alien RNAs.	[50]
Cucumber mosaic virus (CMV)	*N. benthamiana*	PAMs	CMV was artificially injected into *A. thaliana* and *N. benthamiana* through vector pCR01. The pCR01 vector contained F. novicida Cas9 (FnCas9) a codon-optimized protein that is driven by an enhanced 35S promoter and a short-range RNA (sgRNA) guide that is driven by an AtU6 promoter. Complementary oligonucleotides were synthesized based on target gene sequences and were inserted inside the pCR01 vector efficiently to construct 23 corresponding vectors of the pCR01-sgRNA complex.	[51]
Cucumber Vein Yellowing Virus (CVYV), Papaya ring spot virus-W (PRSV-W) and Zucchini yellow mosaic virus (ZYMV)	*Crocus sativus*	*eIF4E*	CMV was artificially injected into *A. thaliana* and *N. benthamiana* through vector pCR01. The Cas9/sgRNAs complex was constructed to target the eIF4E gene in cucumber plants. The sgRNA1 sequence was expected to destroy the intact eIF4E gene, and the sgRNA2 sequence to allow the translation of two-thirds portions of the total protein products. One diploid genome with another single eIF4E gene and homozygous mutant plants were propagated to knock out the expression of the eIF4E gene. The Cas9/sgRNA complex was employed to disrupt the function of the recessive eIF4E gene and thus enhance virus resistance in cucumber.	[52]
Rice tungro spherical virus (RTSV)	*Oryza sativa*	*eIF4G*	Natural resistance to RTSV is a recessive trait that is controlled by a gene, namely, translation initiation factor 4 gamma gene (eIF4G). Mutations that occurred within eIF4G genes were generated utilizing the CRISPR/Cas9 technology to develop new resistance sources in the RTSV-susceptible variety IR64. The final products containing RTSV resistance were found to no longer have the Cas9 sequence under greenhouse conditions.	[53]
Turnip mosaic virus (TuMV)	*A. thaliana*	*eIF4E* and eIF*(iso) 4E*	The CRISPR/Cas9 technology was applied to generate the significant genetic resistance against TuMV in *A. thaliana* plants by deletion of a known host factor (eIF(iso)4E), which was strictly needed for viral existence. Transgenic delivery of the Cas9-sgRNA complex through CRISPR/Cas9 technology has proved its feasibility for the segregation of the transgene originated by induced mutation at the targeted eIF(iso)4E location, initially to generate stable and heritable mutations except for any persistent transgene. This approach is hypothesized as the reason why the recessive gene allele eIF(iso)4E exhibits more durable resistance against TuMV infection; the possible presence of VPg polymorphisms performing via eIF(iso)4E is an independent pathway.	[54]
Tomato yellow leaf curl Sardinia virus (TYLCSV)	*N. benthamiana*	CP and IR	The virus highly conserved a non-nucleotide sequence, which forms a stem-loop-like structure inside the IR sites. This conserved structure was directly involved in the Rep binding site in virus replication and contained some illegal bidirectional gene promoters. The IRs were also strain-specific and associated only with targeted Rep proteins. Authentic TYLCSV-IR-sgRNA was utilized to target the virus TYLCSV. Infectious clones of TYLCSV were injected in *N. benthamiana* plants via agro-infection overexpressing the CRISPR/Cas9 method. The IR target sequences were examined by loss assay assessment of the SspI enzyme, which confirmed the complete absence of recognized indels.	[55]
Tomato yellow leaf curl virus (TYLCV)	*N. benthamiana; Solanum lycopersicum*	CP, IR and Rep sequences	Agrobacterium-mediated transfer DNA (T-DNA) modification was used to express the sgRNAs cassettes by U6-26s promoter and Cas9 protein by CaMV-35S promoter in *N. benthamiana* plants. The U6-sgRNA cassette and Cas9 protein were cloned within a binary vector controlling by the CaMV-35S promoter and transferred in the *N. benthamiana* leaf discs utilizing Agrobacterium tumefaciens. The results proved the ability of CRISPR/Cas9 technology to target the infectious strains of TYLCV in the CP, IR and Rep sequences in transgenic *N. benthamiana* plants.	[27,55,56,57]
Wheat dwarf virus (WDV)	*Hordeum vulgare*	CP, MP, LIR and Rep	To identify the multiple target regions, the WDV genome was mapped for efficient CRISPR/Cas9 target sequences encircling the PAMs motif. Four different target sites were selected that showed no off-target effects and were capable of attacking several viral DNA sequences. The sgRNA WDV1 exhibits the complementarity overlapping in the CP and MP coding regions, sgRNA WDV2 targets the Rep/Rep A coding regions remaining in the N-terminus of the proteins, sgRNA WDV3 targets the LIR region and sgRNA WDV4 targets the Rep protein region and encodes the C-terminus of the protein.	[58]

**Table 3 plants-10-01264-t003:** A list of identified virus resistance genes in 20 food crops, is expected to apply CRISPR/Cas9 to control viral diseases.

SL. No.	Virus in Food Crop	Identified Virus Resistant Gene	Reference
01	African cassava mosaic virus (ACMV)	*CMD1* (recessive resistance gene), *CMD2* (major dominant gene) and *CMD3* (conferring resistance)	[61,62]
02	Banana bunchy top virus (BBTV)	*BBTV DNA-R* and *BBTV DNA-S1*	[63]
03	Barley yellow dwarf virus (BYDV)	*Bdv1, Bdv2*, *Bdv3*, *Bdv4* and *Ryd**2*	[64,65]
04	Bean common mosaic virus (BCMV)	*I* (dominant resistance gene)*, bc**-**u, bc**-**1, bc**-**1^2^, bc**-**2, bc**-**2^2^*, *and bc**-**3* (recessive resistance gene)	[66]
05	Bean yellow mosaic virus (BYMV)	*rym4/5*	[67]
06	Bean golden mosaic virus (BGMV)	*bgm-1* and *bgm-2*	[68,69]
07	Beet yellows virus (BYV)	*III, V* and *VI QTLs*	[70]
08	Grapevine fanleaf virus (GFLV)	*F13, EcoRI* and *Sty**I*	[71]
09	Lettuce mosaic virus (LMV)	*eIF(iso)4G1,**mo11* and *mo12* (recessive resistance gene), Mo2 (dominant resistance gene)	[72]
10	Maize dwarf mosaic virus (MDMV)	*Mdm1*	[73]
11	Maize streak virus (MSV)	*msv1*	[74]
12	Plum pox virus (PPV)	*eIF(iso)4G1, eIF(iso)4E,* *eIFiso4G11, PpDDXL ParP-1 to Par-P-6, ParPMC1 and ParPMC2*	[75]
13	Potato leafroll virus (PLRV)	*Rl_adg_*	[76]
14	Potato virus X (PVX)	*Rx1* and *Rx2*	[77]
15	Potato virus Y (PVY)	*Ry_adg_, Ry_sto_* *, Y-1, pvr1, pvr21, pvr22 + pvr6, pot-1*	[78]
16	Rice tungro bacilliform virus (RTBV)	*RTBV ORF IV* and *RTBV-CP*	[79]
17	Squash leaf curl virus (SLCV)	*slc-2*	[80]
19	Sugarcane mosaic virus (SCMV)	*Scmv1 and Scmv2*	[81]
19	Soybean mosaic virus (SMV)	*Rsv1, Rsv3, Rsv4, Rsv5, Rsv7, Rsv8, Rsv15* and *Rsv20*	[82]
20	Watermelon mosaic virus (WMV)	*Wmv*1551	[83]

## Data Availability

Not applicable.

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
