# Peer review of "Control of Plant Viral Diseases by CRISPR/Cas9: Resistance Mechanisms, Strategies and Challenges in Food Crops"

_plants, 2021, doi:10.3390/plants10071264_

Round 1

Reviewer 1 Report

Reviesed paper "CRISPR/Cas9: Enhancing Resistance against Plant Viruses and Prospects" is very interesting and very important.

Table is perfect but quality of all Figures is very poor. Needs correction.

In Review type papers very important is a meta-analysis. In this paper lack of meta-analysis.

Paper needs major revision.

Author Response

Authors' response: 

On behalf of my co-authors, I want to apologize for the delay in returning a revised version of our manuscript. This delay is partly due to all of us working remotely during the COVID-19 pandemic and reviewing our data analysis. We also want to thank the editor and reviewers for their thoughtful comments, suggestions and recommendations on our manuscript. We carefully considered all of the comments and suggestions and provided the following responses ("See Lines" refers to revised manuscript line numbering), and Yellow lines in the manuscript are the current and new edits:  

Reviewer #1:

Comment 1:

English language and style, extensive editing of English language and style, moderate English changes required, English language and style are fine/minor spell check required, I don't feel qualified to judge about the English language and style.

Authors' responses:

Professional editing services of English language and style has been done throughout the manuscript. A spell check has also been performed. Thank you for this suggestion. Yellow lines in the revised manuscript are edits and new texts.

Comment 2:

The revised paper "CRISPR/Cas9: Enhancing Resistance against Plant Viruses and Prospects" is very interesting and very important.

Authors' responses:

Thank you for recognizing the importance of this virus-Cas9 review topics

Comment 3:

All Figures is very poor. Needs correction.

Author's responses:

The quality figures have been added, and poor-quality figures were removed from this article. See Figures.

Comment 4:

In Review type papers, very important is a meta-analysis. In this paper lack of meta-analysis.

Author's responses:

A comprehensive meta-analysis of CRISPR/Cas9 gene-editing technology is performed. The strategies used to improve crop virus resistance in commercially cultivated crops, specifically the most important field crops, are presented in Table 3. We critically analysed the information coming out from 2015-2021 peer-reviewed journal articles related to Cas9 application in plant viruses.

Comment 5:

Paper needs major revision.

Author's responses:

Major revisions are performed. We carefully addressed all questions and suggestions raised by the reviewers and provided further explanation where necessary. 

Reviewer 2 Report

It is comprehensive review about usage of CRISPR system for increse virus-tolerance.

However, the text does not have lines number what make very difficult to write a precise comments/sugegstions.

please, relöaos text with lines number!

I made some comments, mentioned more, but it is not so easy to mentio them becaise absence of "coordinate system".

Title: please, change: CRISPR is a tool for enhancing resistance.   

„CRISPR/Cas9: Enhancing Resistance against Plant Viruses and Prospects“.

Summary: This sentence is not clear„CRISPR/Cas9 system employed how genome engineering has been used to attack these viruses, which are briefly mentioned“.

Please, make summary more structural and logical.

  1.  

Edition of the grammar is required.

For example: „The development of genetic diversity with enhanced protection to virus genes is an indispensable part of exploiting the principle of virus resistance plant breeding.

  1. Life Cycle of Virus

It will be better write this: „The virus begins to multiply itself in a suitable environment, and replication occurs very fast by the millions.“

Please, adjust another senetnces when neccessary.

Please, provide in the text explanations to the figure 1, namely differences between two cycles and when each of them happens.

Page 5:

Please, give expanation what is „S-genes“.

An alternate ap-proach, such as theaforementioned cis-regulatory component and promoter editing, alters gene expression in lieu of the gene disruption.

Please, edit: „An alternate ap-proach, such as theaforementioned cis-regulatory component and promoter editing, alters gene expression in lieu of the gene disruption“ to „An alternate approach, such as the aforementioned cis-regulatory component and promoter editing, alters gene expression instead of the gene disruption.“.

Please, check other sentensces for typos/grammar.

 Table: please, make better layout: columns 1 and 2 can be less wide, but column 5 more wide.

Author Response

Comment 1:

Extensive editing of English language and style required, moderate English changes required, English language and style are fine/minor spell check required, I don't feel qualified to judge about the English language and style.

Author's responses:

Professional editing of English language and style has been done throughout the manuscript. A spell check has also been performed. Thank you for this suggestion. Yellow lines in the revised manuscript are edits and new texts.

Comment 2:

It is a comprehensive review about usage of CRISPR system for increased virus-tolerance, however, the text does not have lines number what make very difficult to write a precise comments/suggestion, please, reloads text with lines number!

Author's responses:

We are sorry. Now we have added line numbers. We misunderstood the instruction to authors by the journal. Now we can see line numbers on the right side of the manuscript.

Comment 3:

Title: please, change: CRISPR is a tool for enhancing resistance or CRISPR/Cas9: Enhancing Resistance against Plant Viruses and Prospects ".

Authors' responses:

We have changed the title to "CRISPR/Cas9 as a Gene-Editing Tool for Enhancing Viral Disease Resistance in Commercially Cultivated Crops: Recent Advancements, Challenges and Prospects. Thank you for this suggestion. Please see Line # 1-4

Comment 4:

This sentence is not clear "CRISPR/Cas9 system employed how genome engineering has been used to attack these viruses, which are briefly mentioned ".

Authors' responses:

The sentence has been revised as "In this review, CRISPR/Cas9 system and its application in stable crop resistance against destructive plant viruses are briefly described". Please see Line # 31-32

Comment 5:

Please, make summary more structural and logical.

Authors' responses:

We concluded with the key research gaps including 1. The off-targets are among the limitations for large-scale field application of the CRISPR/Cas9 tool in virus-resistant crop development. 2. The impact of the CRISPR/Cas9 methodology on plant physiological traits and potential virus mutants is currently inadequately documented. 3. Global-scale collaboration through open field trial (multi-climate/location) data on Cas9 treated crop plant x virus x environment interactions and subsequent growth and yield performances of stable food crop varieties warrants further attention. Please see Line # 777-785

Comment 6:

Edition of the grammar is required, for example: "The development of genetic diversity with enhanced protection to virus genes is an indispensable part of exploiting the principle of virus resistance plant breeding. "

Authors' responses:

Grammatical errors are attended as "The genetic diversity with enhanced protection to virus genes is an indispensable part of virus resistance". Please see Line # 54-55

Comment 7:

Life Cycle of Virus, It will be better write this: "The virus begins to multiply itself in a suitable environment, and replication occurs very fast by the millions. "

Please, adjust another sentences when necessary.

Authors' responses:

The sentence is revised according to your comment. Please see Line # 92-93

Comment 8:

Please, provide in the text explanations to the figure 1, namely differences between two cycles and when each of them happens.

Authors' responses:

We have added the explanation of the figures as a note. Please see Line # 106-115

Comment 9:

Please, give explanation what is "S-genes ".

Authors' responses:

S-genes are explained/abbreviated to Susceptible (S) genes. Please see Line # 175

Comment 10:

 An alternate approach, such as the cis-regulatory component and promoter editing, alters gene expression in lieu of the gene disruption. Please, edit: "An alternate approach, such as the aforementioned cis-regulatory component and promoter editing, alters gene expression in lieu of the gene disruption "to" An alternate approach, such as the aforementioned cis-regulatory component and promoter editing, alters gene expression instead of the gene disruption. ".

Authors' responses:

The sentence is revised as "An alternate approach, such as cis-regulatory component and promoter editing, is capable of altering gene expression instead of the gene disruption". Please see Line # 176-177

Comments 11: 

Please, check other sentences for typos/grammar.

Authors' responses:

The article is checked with professional editors. Please see the yellow texts in the manuscript. 

Comments 12:

Table: please, make better layout: columns 1 and 2 can be less wide, but column 5 more wide.

Authors' responses

Tables are formatted for a better layout. Please see Tables.

Reviewer 3 Report

This manuscript is well written, nicely presented, and contains a lot of information on the interesting topic of virus resistance in plants through CRISPR/Cas9.

There are serious flaws in the manuscript. Recent papers that very significant in this context are not mentioned, such as the one on CRISPR/Cas9-Mediated Generation of Pathogen-Resistant Tomato against Tomato Yellow Leaf Curl Virus, https://doi.org/10.3390/ijms22041878

A large part of the manuscript covers 28 viruses under the heading of “some commercially important plant viruses”. If one wants to talk of commercially important plant viruses, many more are missing. But what this part of the manuscript really makes it unnecessary is that only a small portion of the list is discussed in the next, main section “Virus resistance via CRISPR/Cas9”, making the reader to wonder about the reason of the 22-virus-species-list. And even so, CRISPR/Cas9 resistance cases are missing e.g. WDV: Kis A, Hamar É, Tholt G, et al. Creating highly efficient resistance against Wheat dwarf virus in barley by employing CRISPR/Cas9 system. Plant Biotechnol J 2019;17(6):1004–6.

There are several recent reviews on the same subject that are more complete and comprehensive, por instance:

https://doi.org/10.1016/j.plaphy.2019.12.022

https://doi.org/10.1094/PHYTO-07-19-0267-IA

doi: 10.3389/fmicb.2020.593700

Author Response

Comment 1:

English language and style, Extensive editing of English language and style required, Moderate English changes required, English language and style are fine/minor spell check required, I don't feel qualified to judge about the English language and style

Authors’ responses:

Professional editing services of English language and style has been done throughout the manuscript. A spell check has also been performed. Thank you for this suggestion. Yellow lines in the revised manuscript are edits and new texts.

Comment 2:

This manuscript is well written, nicely presented, and contains a lot of information on the interesting topic of virus resistance in plants through CRISPR/Cas9.

Authors' responses:

Thank you for recognizing the importance and presentation of this review article.

Comment 3:

There are serious flaws in the manuscript. Recent papers that very significant in this context are not mentioned, such as the one on CRISPR/Cas9-Mediated Generation of Pathogen-Resistant Tomato against Tomato Yellow Leaf Curl Virus, https://doi.org/10.3390/ijms22041878

Authors' responses:

Tomato Yellow Leaf Curl Virus is included in text and reference. Please see line # 894-895

Comment 3:

A large part of the manuscript covers 28 viruses under the heading of "some commercially important plant viruses". If one wants to talk of commercially important plant viruses, many more are missing.

Authors’ responses:

Thank you for this recommendation to add few crops which have been commercially cultivated and considered as the economically important plant viruses across the world. We intended to review major commercial field crops threatened by viruses and the recent development of genome-editing Cas9 tools to combat plant disease control. We discussed a total of 45 viruses, including newly added 17 plant viruses. These lists of viruses are a good representation of commercially cultivated high-value crops worldwide. We also revised the title of Table 2. As "a comprehensive list of forty-five commercially important crop viruses". Our title is now; "CRISPR/Cas9 as a Gene-Editing Tool for Enhancing Viral Dis-ease Resistance in Commercially Cultivated Crops: Recent Advancements, Challenges and Prospects". Please see line # 1-4

Comment 4:

But what this part of the manuscript really makes it unnecessary is that only a small portion of the list is discussed in the next, main section "Virus resistance via CRISPR/Cas9", making the reader to wonder about the reason of the 22-virus-species-list. And even so, CRISPR/Cas9 resistance cases are missing e.g. WDV: Kis A, Hamar É, Tholt G, et al. Creating highly efficient resistance against Wheat dwarf virus in barley by employing CRISPR/Cas9 system. Plant Biotechnol J 2019;17(6):1004–6.

Authors' responses:

We thoroughly read out the article, and the information added in the text and references are listed as [47]. See line # 664-677 & 900-902

  1. There are several recent reviews on the same subject that are more complete and comprehensive, por instance:

https://doi.org/10.1016/j.plaphy.2019.12.022

https://doi.org/10.1094/PHYTO-07-19-0267-IA

doi: 10.3389/fmicb.2020.593700

Authors' responses:

We agree with your perspectives. However, our review focused on the practical application of Cas9 to many high-value crops' viral diseases. Several crops discussed in this review are staple food sources (e.g., wheat, rice, potato) in many countries in the world. Several reviews did not represent a similar subject line. For instance, above, Kalinina et al. (2019) describe the basic principles of CRISPR/Cas systems and how they can only be deployed to model plants. Similarly, Mushtaq et al. (2020) focused on the CRISPR/Cas9 system's significant pitfalls that utilize highly efficient and novel platforms to engineer interference to single and multiple plant RNA viruses. In this sense and point of view, the readers would get a complete set of updated information on field application of Cas9 and subsequent challenges in controlling many (18 crop viruses discussed) devastating crops viruses. These three viruses (CVYV, PRSV-W and ZYMV) were discussed together. See Table 3.

We thoroughly read out these recent articles and added in the text. The articles are cited as [42], [43], [44], [46] and [47]. Revision is highlighted as red mark in references section. See line # 890-902

Round 2

Reviewer 1 Report

Now paper is perfect.

Author Response

Reviewer 1

(x) I don't feel qualified to judge about the English language and style 

Author’s response: English editing services was performed by appropriate professional editors. The text changes marked with yellow colour throughout the manuscript.

Reviewer 2 Report

Thank you for correction, th etext is much better, but some points still require further work.

Line 55: this is obvious statement, I am not sure you need to mention: „Genetic variability is an integral part of resistance plant breeding“.

Line 56: „protection to virus“ is enough, do not need to add „ genes“.

Line 88: „Adsorption occurs between the virus particle and plant-host cell membrane.“- adsorbition can not be between. Please, edit.

Line 248: what is „the long-term resistance against target viruses“? Please, edit.

Line 250: „understanding of S gene’s function enables to recognize the protein locations“ do not directly link with protein location.

Line 260/263; it is an obvious point. Please, edit.

Line 372: „Plant viruses have immense economic importance…“ – plant visuese does not have an economic importance, infection has a negative impact. Please, edit.

Line 422: there is no commercialy cultivated viruses! „CRISPR/Cas9 Strategy Against Commercially Cultivated Crop Plant Viruses“

Line 424:  seuences ??

Line 440-441: please, edit.

Line 475: protein interact through protein-protein interactions? Can they interect by another way?  „Viral protein VPg interacted with diverse members of eIF4E proteins of the cassava 474 family through protein-protein interaction.“

Line 705: „This study found“ – I think it is better to write „This review described“. Find rather link to experimental work.  

Author Response

Reviewer 2

-Moderate English changes required

Author’s response: English editing services was performed by appropriate professional editors. The text changes marked with yellow colour throughout the manuscript.

Comment 1:

Comments and Suggestions for Authors

Thank you for correction, the text is much better, but some points still require further work.

Author’s response: Thank you very much for your kind comments, suggestions and corrections for further improvement of the review.

Comment 2:

Line 55: this is obvious statement, I am not sure you need to mention: „Genetic variability is an integral part of resistance plant breeding“.

Author’s response: In line 55, we have updated the manuscript by removing the sentence “Genetic variability is an integral part of resistance plant breeding” as per the comments of Reviewer 2.

Comment 3:

Line 56: „protection to virus“ is enough, do not need to add „ genes“.

Author’s response: In line 56, “protection to virus genes” changed to “protection to virus”. Texts are highlighted as the yellow mark.

Comment 4:

Line 88: „Adsorption occurs between the virus particle and plant-host cell membrane.“- adsorbition cannot be between. Please, edit.

Author’s response: In line 88, “Adsorption occurs between the virus particle and plant-host cell membrane” changed to “After adsorption of virus to the host cell membrane, a small hole forms……..”. Texts are highlighted as the yellow mark.

Comment 5:

Line 248: what is „the long-term resistance against target viruses“? Please, edit.

Author’s response: In line 191, “the long-term resistance against target viruses” changed to “produce durable virus resistance in different hosts”. Texts are highlighted as the yellow mark.

Comment 6:

Line 250: „understanding of gene’s function enables to recognize the protein locations“ do not directly link with protein location.

Author’s response: In line 192-193, The sentence has been updated as “The functional analysis of susceptibility genes enables us to understand the regulation of gene expression”. Texts are highlighted as the yellow mark.

Comment 7:

Lines 260/263; it is an obvious point. Please, edit.

Author’s response: In line 201-204, These sentences have been rephrased as “it can be assumed that the differences in nucleotide sequences of target gene among the cultivated species and wild varieties are not significant to confer durable resistance to various plant viruses. In this case CRISPR/Cas9 tool can be utilized to modify of a particular gene for stable plant resistance”. Highlighted as yellow mark.

Comment 8:

Line 372: „Plant viruses have immense economic importance…“ – plant virus does not have an economic importance, infection has a negative impact. Please, edit.

Author’s response: In lines 325-327, the sentence has been rephrased as “Plant viruses have immense importance in agriculture because many of them infect vegetable crops, fruits and ornamental plants which deteriorate the productivity and the quality of the plant products. Texts are highlighted as the yellow mark.

Comment 9:

Line 422: there is no commercialy cultivated viruses! „CRISPR/Cas9 Strategy Against Commercially Cultivated Crop Plant Viruses“

Author’s response: In line 361, “CRISPR/Cas9 Strategy Against Commercially Cultivated Crop Plant Viruses” changed to “Application of CRISPR/Cas9 in Food Crops”. Texts are highlighted as the yellow mark.

Comment 10:

Line 424:  seuences ??

Author’s response: In lines 363-366, The sentence has been edited as “The gRNAs were designed to target the sequences of BSOLV and eBSOLV via the CRISPR/Cas9 technology. Three gRNA based on their specificity to their target site (targeting sequence S1, S2 and S3 from ORF1, ORF2 and ORF3, respectively) and minimal potential off-targets were introduced into the triploid Musa genome (Gonja Manjaya, AAB)”. Highlighted as yellow mark.

Comment 11:

Line 440-441: please, edit.

Author’s response: In line 380-382, This part has been edited as “CRISPR/Cas9 technology is successfully utilized in engineering resistance to bean yellow dwarf virus (BeYDF). Baltes et al. [57] established a transient assay to detect the activity of Cas9 and sgRNA in N. benthamiana using BeYDV. They used double 35S promoter and AtU6/At7SL RNA polymerase III promoter to express Cas9 and sgRNAs, respectively”. Texts are highlighted as the yellow mark.

Comment 12:

Line 475: protein interact through protein-protein interactions? Can they interect by another way?  „Viral protein VPg interacted with diverse members of eIF4E proteins of the cassava family through protein-protein interaction“

Author’s response: In line 416-418, The sentence has been rephrased as “Cassava brown streak virus (CBSV) is the great threat for cassava production in Africa. For the disease development in the host, CBSV requires the interaction between ‘viral genome‐linked protein (VPg)’ and ‘eukaryotic translation initiation factor 4E (eIF4E) isoforms’ of the host”. Texts are highlighted as the yellow mark.      

Comment 13:

Line 705: „This study found“ – I think it is better to write „This review described“. Find rather link to experimental work.  

Author’s response: In line 594, “This study found” changed to “This review pronounced”. Texts are highlighted as the yellow mark.

Reviewer 3 Report

This revised manuscript has addressed several points that were highlighted in the previous version. However, the changes that have been made do not improve the manuscript. I was surprised to find 22 "some commercially important plant viruses" in the previous version. This information is now expanded as a Table 2., named “ A comprehensive list of forty-five commercially important crop viruses”. I still do not understand the meaning of this information since all of the species are not further treated within the scope of the paper’s title “CRISPR/Cas9 as a Gene-Editing Tool for Enhancing Viral Disease Resistance in Commercially Cultivated Crops: Recent Advancements, Challenges and Prospects”.

In the present version now there is a new Table 1. The authors should check the orthography of the title, since it contains various errors.  The information supplied in this table is 1) not new, but covered in other reviews, 2) uncomplete (So, resistance genes Ty-x against TYLCSV, but what about a simple TYLCV ?), 3) unnecessary (why present a lot of information which further on is not discussed in detail within the scope of this paper?) and 4) full of mistakes: it only took me two cross checks of the references to question the reason and the quality of this table: reference number 56 is used to support resistance genes for LMV, MDMV, MNSV...It is supposed to refer to the paper from Gosavi et al. , which does not describe the viruses nor the genes; reference number 59 is cited to support resistance genes against ToMV, TMoV, TSWV and TYLCSV. This is wrong. Reference number 59 is the paper by  M. E. Ali et al., and deals with tobamovirus resistance genes, so perhaps  with ToMV in this list only. I must say I did not check the remaining info and references, but if I find these errors in the first two double checks, I am afraid that many more are present.

Author Response

Reviewer 3

Moderate English changes required

Author’s response: Appropriate professional editors performed English editing services. The text changes marked with yellow colour throughout the manuscript.

Comments and Suggestions for Authors

This revised manuscript has addressed several points that were highlighted in the previous version. However, the changes that have been made do not improve the manuscript. I was surprised to find 22 "some commercially important plant viruses" in the previous version. This information is now expanded as a Table 2, named “ A comprehensive list of forty-five commercially important crop viruses”.

In the present version now there is a new Table 1. The authors should check the orthography of the title, since it contains various errors.  The information supplied in this table is 1) not new, but covered in other reviews,

Author’s response: 

Thank you so much for providing helpful directions to improve the manuscript. We agree to revise the title, and we did change the title to “Control of Plant Viral Diseases by CRISPR/Cas9: Resistance Mechanisms, Strategies and Challenges in Food Crops”

Accordingly, we have deleted non-food crop species from Tables 1, 2 and 3, and changed the title of these tables. Subsections of the texts and tables are rearranged, aligning with the title. For example, in previous Table 1 moved to under subsection 13—challenges as Table 3 (please see line 591). We considered Table 3 as essential in this manuscript because the listed 20 viruses are virulent to many food crops. However, the resistance genes in respective crops have already been identified, but a long way to go to have successful application of cas9 technology to control these viral pathogens in many staple food crops (potato, bean, barley, etc.).

The previous version described resistance strategies of some non-food crops, e.g., fibre (cotton) and narcotics (tobacco) crop species excluded from the manuscript. Moreover, in this manuscript, we intend to focus on “resistance mechanisms, strategies, and current challenges” with cas9, specific to food crop species. That is why we highlighted our title with these key words. Comments and Suggestions for Authors

Comment 1:

2) Incomplete (So, resistance genes Ty-x against TYLCSV, but what about a simple TYLCV?),

Author’s response: Both have been removed from the current Table 3.

3) Unnecessary (why present a lot of information which further on is not discussed in detail within the scope of this paper?) and

Author’s response: Thank you for these valuable suggestions. We have excluded that information that has not been discussed within the scope of the paper's objectives. For example, subsection 4 (Gene-editing using Cas9 Protein) (in the last version, line 136),    and 12 (Genetics of Plant Virus-Resistance) (in the last version, line 342) have been excluded from the current manuscript as the contents of the sections are not reasonably fit with our objectives.

Comment 1:

4) full of mistakes: it only took me two cross checks of the references to question the reason and the quality of this table: reference number 56 is used to support resistance genes for LMV, MDMV, MNSV...It is supposed to refer to the paper from Gosavi et al. , which does not describe the viruses nor the genes; reference number 59 is cited to support resistance genes against ToMV, TMoV, TSWV and TYLCSV. This is wrong. Reference number 59 is the paper by  M. E. Ali et al., and deals with tobamovirus resistance genes, so perhaps  with ToMV in this list only. I must say I did not check the remaining info and references, but if I find these errors in the first two double checks, I am afraid that many more are present.

Author’s response:

We are sorry for these mistakes. While editing, the format of the manuscript was changed unintentionally. We cross checked and ensuring all references (references no 74-96) are correctly cited.  Highlighted as yellow mark in Table 3.